# LOCALIZED TEXT-TO-IMAGE GENERATION FOR FREE VIA CROSS ATTENTION CONTROL

## ABSTRACT

Despite the tremendous success in text-to-image generative models, *localized* text-to-image generation (that is, generating objects or features at specific locations in an image while maintaining a consistent overall generation) still requires either explicit training or substantial additional inference time. In this work, we show that localized generation can be achieved by simply controlling cross attention maps during inference. With no additional training, model architecture modification or inference time, our proposed cross attention control (CAC) provides *new* open-vocabulary localization abilities to standard text-to-image models. CAC also enhances models that are already trained *for* localized generation when deployed at inference time. Furthermore, to assess localized text-to-image generation performance *automatically*, we develop a standardized suite of evaluations using large pretrained recognition models. Our experiments show that CAC improves localized generation performance with various types of location information ranging from bounding boxes to semantic segmentation maps, and enhances the compositional capability of state-of-the-art text-to-image generative models.

## 1 INTRODUCTION

Text-to-image generative models have shown strong performance in recent years: models like Stable Diffusion (Rombach et al., 2021) and Dall-E (Ramesh et al., 2021) are capable of generating high quality and diverse images from arbitrary text prompts. However, a significant challenge faced by these models is that they rely *solely* on text prompts alone for content control over the generation process, which is inadequate for many applications. Specifically, one of the most intuitive and user-friendly ways to exert control over the generation is to provide *localization information*, which guides the models on where to generate specific elements within the image. Unfortunately, current pretrained models face limitations in their capability to perform localized generation. These limitations arise not only from their inability to incorporate location information as input but also from the inherent difficulties associated with compositionality, which is a known challenge for many multimodal foundation models (Thrush et al., 2022).

Existing methods addressing this issue typically fall into three main categories: training entirely new models (Park et al., 2019; Isola et al., 2017), fine-tuning existing models with additional components such as task-specific encoders (Li et al., 2023), or strategically combining multiple samples into one (Liu et al., 2022; Bar-Tal et al., 2023). All of these approaches often demand a substantial amount of training data, resources, and/or extended inference time, rendering them impractical for real-life applications due to their time and resource-intensive nature. On the other hand, in a separate but related line of work, Hertz et al. (2022) proposed Prompt-to-Prompt Image Editing, which edits generated images based on modified text prompts by manipulating cross attention maps in text-to-image generative models. Notably, this work also shows that cross attention layers play a pivotal role in controlling the spatial layout of generated objects associated with specific phrases in the prompts.

In this work, we propose to use cross attention control (CAC) to provide pretrained text-to-image models with better open-vocabulary localization abilities. As illustrated in Figure 1, given a caption and localization information, such as bounding boxes and semantic segmentation maps, along with their corresponding text descriptions, we first construct a new text input by concatenating the caption and all prompts associated with the location information. We then compute the cross attention maps

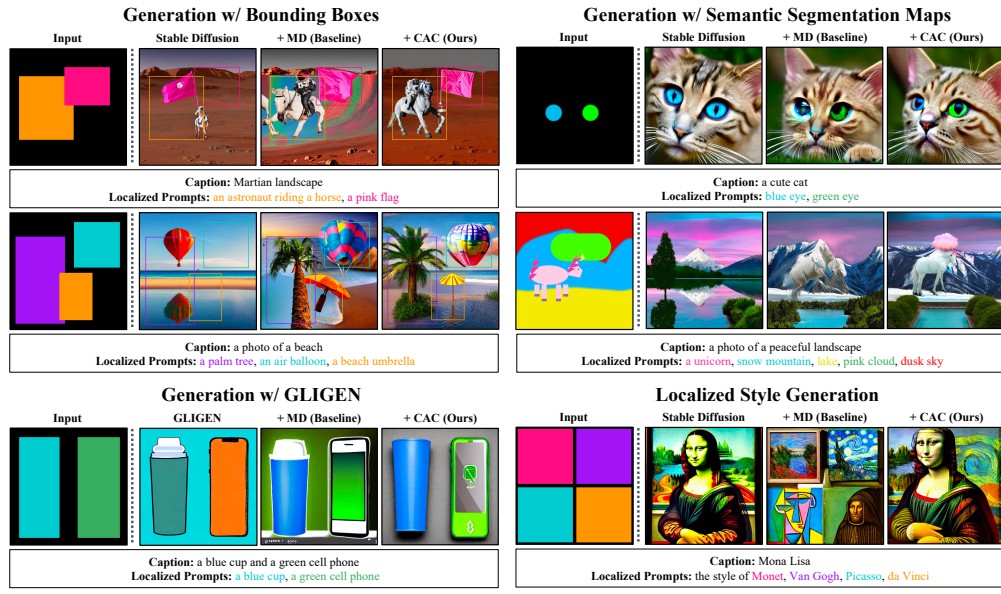

Figure 1: CAC as a plugin to existing methods for localized text-to-image generation. CAC improves upon diverse types of localization (bounding boxes, semantic segmentation maps and localized styles) with different base models (Stable Diffusion and GLIGEN). [2]

from this new text prompt and apply localization constraints to the cross attention maps according to the localization information. Our method does not require any additional training or model architecture modification like designing task-specific encoders. It also does not impose any language restrictions such as using a fixed set of vocabulary or a language parser. Moreover, it is highly portable and can be easily integrated into a single forward pass in any cross attention based text-to-image generation framework with only a few lines of code, thus demanding no extra inference time.

We develop a standardized suite of evaluation metrics for localized text-to-image generation tasks using off-the-shelf large pretrained recognition models (Jocher et al., 2023; Chen et al., 2023; Kirillov et al., 2023; Li* et al., 2022). We apply CAC to various state-of-the-art baseline text-to-image generative models and experiment with different forms of localization information including bounding boxes and semantic segmentation maps. We demonstrate that CAC endows pretrained standard text-to-image models with new localized generation abilities, and furthermore, improves upon models specifically trained for localized generation. In addition, we show that with simple heuristics that spatially separate the components within text prompts, our method can significantly improve the compositional ability of text-to-image generative models.

## 2 RELATED WORKS

As the quality of machine generated images drastically improves over the last decade (Kingma & Welling, 2022; Goodfellow et al., 2014; Sohl-Dickstein et al., 2015; Ho et al., 2020), innovations for enhancing controllability over these models also rapidly developed (Mirza & Osindero, 2014; Dhariwal & Nichol, 2021; Karras et al., 2018; Meng et al., 2022). Recently, text-to-image generative models have shown strong performance (Ramesh et al., 2021; Saharia et al., 2022; Kang et al., 2023; Nichol et al., 2022), many of which leverage cross attention layers to communicate between modalities. Among these models, Stable Diffusion (Rombach et al., 2021) has gained a lot of popularity due to its open source availability. Based on the controllability provided by the text prompts, Hertz et al. (2022) propose an image editing method that manipulates the outputs of the model by modifying the text prompts and controlling the cross attention maps. As applications of these models emerge, however, a significant drawback becomes quickly notable: these models rely solely on text prompts for content control, which is insufficient for many applications scenarios and inaccurate due to the inability to handle compositionality (Thrush et al., 2022).

To tackle the compositionality problem, Composable Diffusion (Liu et al., 2022) interpret diffusion models as energy-based models and explicitly combines the energy functions for each component

---

[2]All shades of pink in the middle right example correspond to the prompt "unicorn".

in the text prompts. StrutureDiffusion (Feng et al., 2022) improves the compositionality ability by incorporating a linguistic parser into the inference time and separately calculate the cross attention for each noun phrase. Stable Diffusion 2.1 improves upon Stable Diffusion 1.4 with a better text encoder. To tackle the insufficient controllability problem, recent models have explored ways to generate images based on location information like bounding boxes (Zhao et al., 2019; Sun & Wu, 2019; Balaji et al., 2023) and semantic segmentation maps (Park et al., 2019; Isola et al., 2017; Xue et al., 2023). In particular, MultiDiffusion (Bar-Tal et al., 2023) solves for an optimization task that binds different regions of the images together based on the localization information provided by the users. GLIGEN (Li et al., 2023) extends the frozen pretrained Stable Diffusion model by adding a set of gated self attention layers.

While many of these methods provide quality results, most of them are highly costly. For example, Park et al. (2019); Isola et al. (2017) need to train entirely new models. Models like GLIGEN require higher-memory GPUs and additional data to train new task-specific layers for different localization information. Training-free methods including Composable Diffusion and MultiDiffusion require $\mathcal{O}(mT)$ inference time where $m$ is the number of objects or features and $T$ is the inference time of the original pretrained models. StructureDiffusion does not impose an extra cost on the pretrained models, but it requires a pre-selected language parser and cannot handle localization information. In contrast to prior work, we propose an approach to solve localized generation problem with no extra cost: our method does not demand extra training, model architecture modification, additional inference time, or any other language restrictions, such as a fixed set of vocabulary or a parser.

## 3 METHOD

### 3.1 PROBLEM SETUP

The goal of this work is to perform localized text-to-image generation given pretrained text-to-image generative models. The localization information provided by the users should consist of text phrases that describe the contents and the constrained spatial locations associated with these contents in the image space. Common location information includes bounding boxes and semantic segmentation maps. Moreover, we aim at performing this task with (1) no additional training or finetuning (2) no model architecture modification and (3) no extra inference time (4) no further limitation on the input text space from the original model. The provided pretrained models can either be trained with localization information, or solely trained with the text-image pairs.

Formally, given a pretrained text-to-image generative model $p_\theta$, a length $n_0$ text prompt $y_0 \in \mathcal{Y}^{n_0}$ and a set of localization information $g = \{g_i\}_{i=1}^m$, our goal is to generate an image $x \in \mathcal{X} \subset \mathbb{R}^{C \times H \times W} \sim p_\theta(x|y_0, g)$ that is visually consistent with the overall textual description provided in $y_0$ and the localized description provided in $g$. Here $\mathcal{Y}$ represents the vocabulary space of the text prompt, $C, H, W$ are the dimensionalities of the output images, and for each $i \in \{1, \cdots, m\}$, $g_i = (y_i, b_i) \in \mathcal{Y}^{n_i} \times [0, 1]^{H \times W}$ where $y_i$ is the textual description of the $i$-th localized region and $b_i$ is the spatial constraint mask corresponding to that description. The pretrained model $p_\theta$ can either sample from $p(x|y_0)$ or $p(x|y_0, g)$. We assume the pretrained models use cross attention mechanism, which we will discuss in the following sections, for the text-image conditioning.

### 3.2 TEXT-TO-IMAGE GENERATION WITH CROSS ATTENTION

State-of-the-art text-to-image generative models achieve their success with cross attention mechanism. Due to the open source availability, we choose Stable Diffusion (Rombach et al., 2021) as the backbone model and we will discuss our method based on its formulation. However, our method can also be applied to other cross attention based diffusion models such as Imagen (Saharia et al., 2022) and GANs such as GigaGAN (Kang et al., 2023).

For the task of sampling from $x \in p_\theta(x|y_0)$ where $x \in \mathcal{X}, y_0 \in \mathcal{Y}^n$, a cross attention layer $l$ in $p_\theta$ receives an encoded text prompt $e_{y_0} \in \mathbb{R}^{n_0 \times d_e}$ and an intermediate sample $z^{(<l)} \in \mathbb{R}^{(H^{(l)} \times W^{(l)}) \times C^{(l)}}$ that has been processed by previous layers in the network and previous diffusion timesteps as its inputs. $n_0, d_e$ are the text length and text embedding dimension, and $C^{(l)}, H^{(l)}, W^{(l)}$ represent the perceptive dimensions of layer $l$, which can be different from $C', H', W'$ because of the U-Net structure. We then project $z^{(<l)}$ into a query matrix $Q^{(l)} = l_Q(z^{(<l)}) \in \mathbb{R}^{h \times (H^{(l)} \times W^{(l)}) \times d}$ and $e_{y_0}$

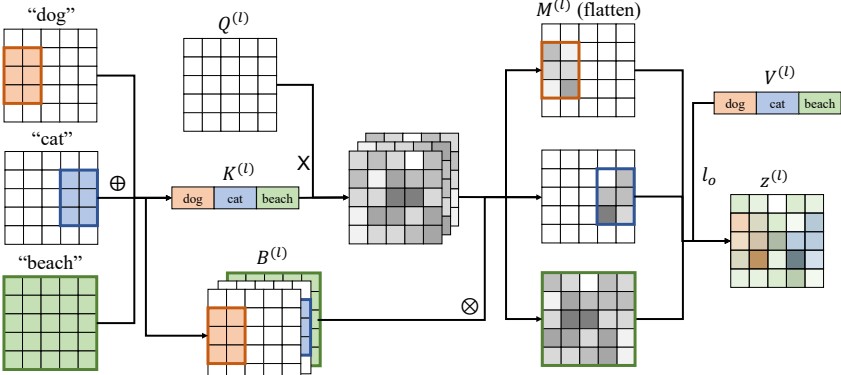

Figure 2: The illustration of CAC for localized generation. CAC uses localized text descriptions and spatial constraints to manipulate the cross attention maps.

into a key matrix and a value matrix $K_0^{(l)} = l_K(e_{y_0}) \in \mathbb{R}^{h \times n_0 \times d}, V_0^{(l)} = l_V(e_{y_0}) \in \mathbb{R}^{h \times n_0 \times d_v}$. $h$ is the number of heads for multihead attention, $d$ is the feature projection dimension of query and key and $d_v$ is that of the value. The cross attention map at layer $l$ is then calculated to be:

$$M_0^{(l)} = \text{Softmax}(\frac{Q^{(l)} K_0^{(l)^\top}}{\sqrt{d}}) \in \mathbb{R}^{h \times (H^{(l)} \times W^{(l)}) \times n_0}. \tag{1}$$

$z^{(l)} = l_O(M_0^{(l)} V_0^{(l)}) \in \mathbb{R}^{(H^{(l)} \times W^{(l)}) \times C^{(l)}}$ is the output of $l$ where $l_O$ is another linear projection.

We can interpret the each entry $M_{0,(r,j,k)}^{(l)}$ in $M_0^{(l)}$ as the extent of attention the $r$-th head pays to the $k$-th token in $y_0$ when generating the $j$-th pixel block in the image. The layer output is the weighted average of the value features, where the weights are assigned by the attention maps from all heads.

## 3.3 CROSS ATTENTION CONTROL (CAC) FOR LOCALIZED GENERATION

Each localization information pair $g_i = (y_i, b_i)$ indicates that the model should generate contents that can be described by text prompt $y_i$ at pixel locations where $b_i > 0$. Therefore, based on the previous interpretation and discovery, the $(r, j, k)$-th element in the attention map should only receive attention from the $k$-th token in $y_i$ if the $j$-th entry $b_{i,(j)}$ in the spatial constraint mask $b_i$ is positive.

As a result, we first interpolate the original location mask $b_i$ to obtain $b_i^{(l)} \in H^{(l)} \times W^{(l)}$ that match dimentionality of the perceptive field of layer $l$. Let $B_i^{(l)} \in \mathbb{R}^{h \times (H^{(l)} \times W^{(l)}) \times n}$ denote the flattened and broadcasted location mask constructed from $b_i^{(l)}$ and $K_i^{(l)} = l_K(e_{y_i}) \in \mathbb{R}^{h \times n_i \times d}$ denote the key matrix calculated from $y_i$, we can extend Equation 1 to have:

$$M_i^{(l)} = \text{Softmax}(\frac{Q^{(l)} K_i^{(l)^\top}}{\sqrt{d}}) \odot B_i^{(l)} \in \mathbb{R}^{h \times (H^{(l)} \times W^{(l)}) \times n_i}. \tag{2}$$

The remaining question is: how should we combine the $m + 1$ attentions maps $\{M_i^{(l)}\}_{i=0}^m$[3]? One intuitive attempt is to calculate the average map $\overline{M^{(l)}} = \frac{1}{m+1} \sum_{i=1}^m M_i^{(l)}$. However, it is unclear what is the "average" value matrix corresponding to this attention map. Another attempt is to separately calculate the matrices $M_i^{(l)} V_i^{(l)}$ where $V_i^{(l)} = l_V(e_{y_i})$, and then calculate the average output matrix $\frac{1}{m+1} \sum_{i=0}^m l_O(M_i^{(l)} V_i^{(l)})$ or $\frac{1}{m+1} l_O(\sum_{i=0}^m M_i^{(l)} V_i^{(l)})$ as the output of the layer. This attempt resembles StructureDiffusion proposed by Feng et al. (2022). While it works well for their standard text-to-image generation task, very sparse attention maps rendered by localization information associated with small objects in our setting can lead to unexpected behaviors.

This question can be much easier to answer if $y_i$ is a substring of $y_0$ for all $i = 1, \cdots, m$: for instance, if a user wants to generate "a photo of a dining room with cups on a dining table" and provides bounding boxes for the "cups" and the "dining table", then we can directly mask the parts of the

---

[3]We define $b_0$ to be an all-one matrix to calculate $M_0^{(l)}$ and all text prompts are padded to the same length.

attention map for the caption (i.e. $M_0^{(l)}$) that are associated with the tokens for "cups" and "dining table" using the location information. Formally, suppose $y_i$ corresponds to the $j_i$-th token to the $(j_i + n_i)$-th token in $y_0$, then we can directly calculate:

$$M_i^{(l)} = M_0^{(l)} \odot B_{i,(j_i:j_i+n_i)}^{(l)}, \tag{3}$$

where $B_{i,(j_i:j_i+n_i)}^{(l)}$ is the mask where $b_i^{(l)}$ is only broadcasted to the $j_i$-th to $(j_i + n_i)$-th submatrices in the third dimension while keeping the rest of the elements all zeros. Then we can calculate $z^{(l)} = l_O((\sum_{i=0}^m M_i^{(l)})V_0^{(l)})$.

However, this assumption may not hold all the time. For example, the user can request to generate "a photo of a dining room" without describing all the details of the scene, but they can still specify the locations of the "cups" and the "dining table" with bounding boxes without mentioning them in the caption. Therefore, to apply this method to all inputs without this assumption, we construct a new text prompt by concatenating all input prompts:

$$y = y_0 \oplus y_1 \oplus \cdots \oplus y_m \tag{4}$$

where $\oplus$ denotes concatenation. We keep all the special tokens from encoding and pad the resulting prompt after concatenation. Similar to the text prompts, we also concatenate all masks to create:

$$B^{(l)} = B_0^{(l)} \oplus B_1^{(l)} \oplus \cdots \oplus B_m^{(l)} \in \mathbb{R}^{h \times (H^{(l)} \times W^{(l)}) \times (\sum_{i=0}^m n_i)} \tag{5}$$

We use all-one matrices as the location masks for the caption $y_0$ and the special tokens in practice.

Similar to Prompt-to-Prompt Editing (Hertz et al., 2022), we can also apply a separate set of weights $\lambda \in \mathbb{R}^{\sum_{i=0}^m n_i}$ to the attention maps to adjust the effects of each token has to the resulting generation. With $K^{(l)} = l_K(e_y) \in \mathbb{R}^{h \times (\sum_{i=0}^m n_i) \times d}$, we can calculate the aggregated attention map as

$$M^{(l)} = \lambda \text{Softmax}\left(\frac{Q^{(l)} K^{(l)\top}}{\sqrt{d}}\right) \odot B^{(l)} \in \mathbb{R}^{h \times (H^{(l)} \times W^{(l)})(\sum_{i=0}^m n_i)}. \tag{6}$$

Finally, the output of the layer can be computed as $z^{(l)} = l_O(M^{(l)} V^{(l)})$ and our framework is illustrated in Figure 2.

Our method only changes the forward pass of the pretrained model at sampling time and thus does not require any further training or model architecture modifications, and it does not demand any other language restrictions or priors such as a fixed set of vocabulary or a language parser. It is also well packaged in the original optimized transformer framework and therefore requires no additional inference time. Because of the minimal assumption, our method is an open-vocabulary plugin for all text-to-image generative models that use cross attention for textual guidance at no extra cost.

## 3.4 INCORPORATING SELF ATTENTION CONTROL

In addition to cross attention, self attention layers are also essential for many text-to-image generative models to produce coherent spatial layouts. Hertz et al. (2022) also found that in addition to cross attention control, applying self attention control to a small portion of the diffusion process can further help provide consistent geometry and color palettes. While self attention control is trivial in editing, it becomes complicated in our setting since location information for different localization prompts can overlap with each other, resulting conflicting signals at the overlapping pixels and ambiguous masks.

One approach to incorporate self attention control is to separately optimize each region according to different localization prompts before binding all regions together. When applying both self attention control and cross attention control to all diffusion steps, the solution to this optimization problem can be roughly reduced to MultiDiffusion (Bar-Tal et al., 2023). As a result, we can first apply MultiDiffusion to a small portion of the diffusion process, and then perform cross attention controlled diffusion as described in 3.3 to the rest of the diffusion timesteps to obtain the desired effect. We can also use models like GLIGEN (Li et al., 2023) that are finetuned on localization information to provide learned self attention control.

Notice that in this case our method is considered a plugin for MultiDiffusion and GLIGEN to provide better localization ability, and it still does not add extra cost to the two algorithms.

Figure 3: Illustration of generated images based on COCO bounding boxes.

# 4 EXPERIMENTS

## 4.1 BASELINES

We select six Stable Diffusion based methods as the baselines to our work. We categorize these baselines into three categories based on the cost to use them: (1) methods that add **no extra cost** to the pretrained model which includes we choose two versions of Stable Diffusion (SD), 1.4 and 2.1, as well as StrutureDiffusion (Feng et al., 2022) and Paint with Words (PWW) (Balaji et al., 2023), (2) methods that **increase inference time** which includes Composable Diffusion (Liu et al., 2022) and MultiDiffusion (MD) (Bar-Tal et al., 2023), and (3) methods that **require additional training** for which we choose GLIGEN (Li et al., 2023) and FreestyleNet (Xue et al., 2023) as the baseline. Methodologies of the baselines are discussed in Section 2. In the next sections, we group the results by the tree categories we introduce in this section. We include the implementation details in the appendix.

## 4.2 LOCALIZED TEXT-TO-IMAGE GENERATION

As mentioned in the introduction, previous works that tackle similar tasks, especially the ones with open vocabulary setting, usually rely on qualitative results and human judgements for evaluations. Due to human involvement in these evaluations, they tend to be expensive and inefficient, and are usually not scalable and prone to high variance. On the other hand, foundation models developed for many general purpose recognition tasks have shown great potential in supporting human to streamline various language and vision tasks. Here we investigate ways to integrate several different off-the-shelf large pretrained models for scalable automatic metrics for localized and compositional text-to-image generation. In each section below, we show that these large pretrained models have the ability to reflect correctly on the relative performance among the generations with different types of input information and agree with qualitative results and human evaluations.

### 4.2.1 GENERATING WITH BOUNDING BOXES

**Experiment Setting and Dataset**    We use the validation set of COCO2017 (Lin et al., 2015) to perform bounding box based generation experiments. Each data point contains a caption for the overall scene and a set of bounding boxes each associated with a class label. Following the settings of Bar-Tal et al. (2023), we create the pseudo text prompt with the class name for each bounding box and filter out 1095 examples with 2 to 4 non-human objects that are larger than 5% of the image area. For models that are unable to take location information as inputs, we create pseudo text prompts by the format "¡caption¿ with ¡object1¿, ¡object2¿, ...". In addition to all the baselines mentioned above, we also test the ability of MultiDiffusion as a plugin to models other than Stable Diffusion to compare with our method.

**Evaluation Metrics**    We evaluate the generated images by (1) how close the generated image resembles a real COCO image (fidelity), (2) how consistent the generations are with the bounding boxes (controllability) and (3) how long it takes for the model to generate an image (inference time). We use Kernel Inception Score (KID) (Bińkowski et al., 2018) to evaluate fidelity. For controllability, we use an YOLOv8 model Jocher et al. (2023) trained on the COCO dataset to predict the bounding boxes in the generated images and then calculate the precision (P), recall (R), mAP50 and mAP50-95 with the ground truth boxes. We use the default thresholds for all metrics. We also report the average inference time with a 50-step sampler on one NVIDIA Tesla V100 machine.

**Results**    Table 1 shows the quantitative results of generation with bounding boxes. As we can observe, CAC improves the localized generation ability for all models we apply it to. As a plugin, CAC also does not substantially increase the inference time compared to MultiDiffusion. In Figure 3,

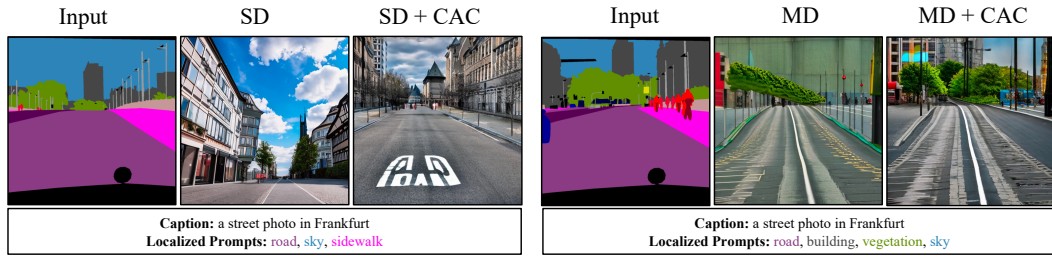

Figure 4: Illustration of different approaches generating images via Cityscapes segmentation maps.

| Method | KID ↓ | P ↑ | R ↑ | mAP50 ↑ | mAP50-95 ↑ | Time (s) ↓ |
|---|---|---|---|---|---|---|
| Ground Truth | - | 0.6010 | 0.5680 | 0.6380 | 0.5460 | - |
| Stable Diffusion (SD) 1.4 | 0.00697 | 0.1120 | 0.0968 | 0.0452 | 0.0146 | 10.19 |
| Stable Diffusion (SD) 2.1 | 0.00733 | 0.0944 | 0.1050 | 0.0588 | 0.0178 | 9.20 |
| StructureDiffusion | **0.00654** | 0.0810 | 0.1070 | 0.0462 | 0.0147 | **8.91** |
| Paint With Words (PWW) | 0.00803 | 0.1780 | 0.1230 | 0.0945 | 0.0339 | 13.43 |
| **SD 2.1 + CAC (Ours)** | 0.00786 | **0.2570** | **0.1990** | **0.1650** | **0.0500** | 9.28 |
| Composable Diffusion | 0.01174 | 0.1560 | 0.0852 | 0.0534 | 0.0165 | 32.75 |
| MultiDiffusion (MD) | 0.01189 | **0.3790** | 0.2820 | 0.2570 | 0.1090 | 27.60 |
| **MD + CAC (Ours)** | **0.00988** | **0.3790** | **0.3050** | **0.2930** | **0.1340** | **16.44** |
| FreestyleNet | 0.00923 | 0.2950 | 0.2610 | 0.1930 | 0.1080 | **11.71** |
| GLIGEN | 0.00691 | 0.7380 | 0.6280 | 0.6740 | 0.4670 | 27.60 |
| GLIGEN + MD | **0.00679** | 0.6940 | 0.6530 | 0.6800 | 0.4690 | 101.15 |
| **GLIGEN + CAC (Ours)** | 0.00708 | **0.7970** | **0.7000** | **0.7810** | **0.5760** | 27.85 |

Table 1: Experiment results with bounding box information.

we illustrate a few qualitative examples that demonstrate the effect of CAC in Stable Diffusion (SD), MultiDiffusion (MD) and GLIGEN. For models that do not have localization ability like SD, CAC provides the new ability to generate contents based on the location information. For models that have localization ability such as MD and GLIGEN, CAC improves the generation by making the generated objects and features more recognizable. Notice that the performance is still strongly influenced by the base model. For example, both SD and SD+CAC generation are missing the "woman" in the caption.

### 4.2.2 Generating with Semantic Segmentation Maps

**Experiment Setting and Dataset**  We use the validation set of Cityscapes (Cordts et al., 2016) dataset for semantic segmentation map based generation. The dataset consists of 500 street photos taken in three cities and the pixel level semantic labeling from 30 predefined classes. We generate a pseudo caption for each image with the format "a street photo in ¡city¿" where ¡city¿ indicates the city where the picture was taken. We also produce pseudo prompts for each semantic segmentation mask associated with each class by using the class name, and we use the same format to create prompts for models without localization ability. We also filtered all classes that occupy less than 5% of the image. We center crop and resize each map to $512 \times 512$ in order to match the dimensionality of the models. We omit the experiments related to GLIGEN due to the unavailability of their pretrained models on Cityscapes.

**Evaluation Metric**  We also evaluate the model performance based on fidelity, controllability and inference time in this experiment. We use KID again to evaluate the fidelity. Since the divergence between data distribution of Cityscapes images and the generated images is substantial according to the KID results in Table 2, semantic segmentation models that are only trained on Cityscapes will not work well in our setting due to the data distribution shift. As a result, we use Semantic Segment Anything (SSA) (Chen et al., 2023), which is a general purpose open-vocabulary model that leverages Segment Anything Model (SAM) (Kirillov et al., 2023) for semantic segmentation. We report the mean IoU (mIoU) score and mean accuracy (mACC) calculated from all classes, and all pixel accuracy (aACC).

| Method | KID ↓ | aACC ↑ | mIoU ↑ | mACC ↑ |
|---|---|---|---|---|
| Ground Truth | - | 91.60 | 60.79 | 71.63 |
| Stable Diffusion 1.4 | 0.128 | 34.85 | 4.54 | 8.83 |
| Stable Diffusion 2.1 | **0.113** | 36.37 | 4.55 | 8.58 |
| StructureDiffusion | 0.132 | 34.99 | 4.60 | 8.82 |
| **SD 2.1 + CAC (Ours)** | 0.115 | **51.29** | **8.20** | **13.20** |
| Composable Diffusion | 0.166 | 47.03 | 6.13 | 10.05 |
| MultiDiffusion | 0.151 | **47.52** | 8.60 | 13.39 |
| **MD + CAC (Ours)** | **0.145** | 46.13 | **8.61** | **13.66** |

Table 2: Experiment results with semantic segmentation information.

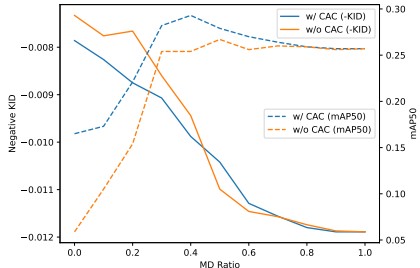

Figure 5: Ablation study on the fideility-controllability tradeoff with and without CAC.

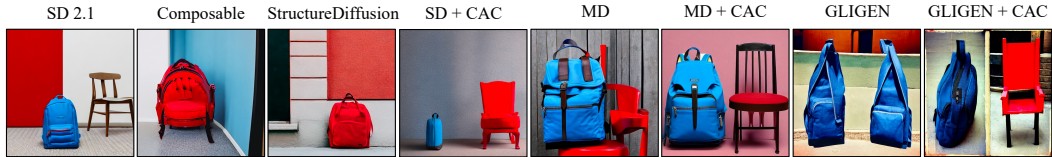

SD 2.1    Composable    StructureDiffusion    SD + CAC    MD    MD + CAC    GLIGEN    GLIGEN + CAC

Figure 6: Different generations with caption "a blue backpack and a red chair" from CC-500 dataset.

**Results**   Table 2 and Figure 4 demonstrate the quantitative and qualitative results for the task of generation with semantic segmentation maps. Although the SSA model can achieve very high accuracy and IoU score for ground truth images, there is still a gap between performance on generated images and real images. Nevertheless, we still find the quantitative results resemble the relative performance of qualitative examination. In particular, like the previous experiments with bounding box information, our method can also provide additional localization ability with semantic segmentation maps. For localized method MD, the generated parts are usually well separated by not coherent, and CAC is able to create more consistent images compared to MD.

### 4.2.3 GENERATING WITH COMPOSITIONAL PROMPTS

The challenge of localized generation stems not only from the incorporation of location information but also from the inherent difficulty of compositional generation. Studies in Thrush et al. (2022) show that compositionality, which is the task of creating complex content from combinations of simpler features and objects, is still an extremely challenging concept even in the era of large pretrained models. Liu et al. (2022); Feng et al. (2022) also find that Stable Diffusion does not exempt from encountering this difficulty - different objects and the relationships between objects and attributes are frequently misrepresented or entirely missing when generating complex scenes. Hence, in this section, we also investigate how our method can improve the performance of compositional generation.

**Experiment Setting and Dataset**   We use the CC-500 dataset (Feng et al., 2022) for this task. CC-500 consists of text prompts with the format "a ¡color 1¿ ¡object 1¿ and a ¡color 2¿ ¡object 2¿". For methods that require location inputs, we apply a simple heuristic where the method will generate the first object on the left hand side of the image and the second object on the right hand side.

**Evaluation Metric**   In this task we mainly evaluate the accuracy of the generation. Following Feng et al. (2022), we categorize each generation into three categories: generations that have incorrect or missing objects, generations that have the correct objects but the wrong colors, and the generations that have the correct objects with the correct colors. Similar to Li et al. (2023); Feng et al. (2022), we choose GLIP (Li* et al., 2022), the semantic rich open-vocabulary object detector to produce object and color detection results. We report the mean percentage of GLIP detection (mGLIP) with confidence threshold from 0.6 to 0.8 with 0.05 step size. Considering the difficulty in recognizing compositional objects, we also perform human evaluation on Amazon Mechanical Turk (mTurk) to verify the automatic results. Details about the human evaluation is in the appendix.

**Results**   Table 3 shows the quantitative results by automatic metrics and human evaluation. Even though there are some discrepancies between machine recognizability and human recognizability, both automatic evaluation and human annotators agree on the relative performance of the models

| Method | Missing/Incorrect Object(s) ↓ | | Correct Objects but Wrong Color(s) ↓ | | Correct Objects & Correct Colors ↑ | | Time↓ |
|---|---|---|---|---|---|---|---|
| | mGLIP | MTurk | mGLIP | MTurk | mGLIP | MTurk | |
| Stable Diffusion (SD) 1.4 | 55.37% | 23.38% | 13.89% | 46.99% | 30.74% | 29.63% | 10.62 |
| Stable Diffusion (SD) 2.1 | 33.56% | 28.24% | 16.53% | 26.62% | 49.91% | 44.91% | 9.58 |
| StructureDiffusion | 52.64% | 25.46% | 11.39% | 40.28% | 35.97% | 34.26% | **8.37** |
| Paint With Words (PWW) | 46.44% | 31.25% | 13.84% | 31.48% | 39.72% | 37.04% | 13.32 |
| **SD 2.1 + CAC (Ours)** | **20.09%** | **7.41%** | **12.64%** | **11.34%** | **67.27%** | **81.25%** | 9.36 |
| Composable Diffusion | 51.25% | 21.99% | 13.94% | 34.03% | 34.81% | 43.75% | 35.78 |
| MultiDiffusion (MD) | 20.05% | 10.88% | 16.48% | 9.72% | 63.47% | 79.40% | 22.33 |
| **MD + CAC (Ours)** | **18.33%** | **7.41%** | **12.08%** | **4.40%** | **69.58%** | **87.96%** | 12.77 |
| FreestyleNet | 65.88% | 36.34% | 12.22% | 37.50% | 21.90% | 25.93% | **11.84** |
| GLIGEN | 47.73% | 43.29% | 11.34% | 29.17% | 40.93% | 27.55% | 27.62 |
| **GLIGEN + CAC (Ours)** | **31.11%** | **8.80%** | **8.98%** | **15.05%** | **59.91%** | **76.16%** | 27.94 |

Table 3: Experiment results on CC-500 dataset. The row with the highest "Correct Objects & Correct Colors" rate is highlighted in each category.

and concur that our method significantly improves the compositional generation capability for all models. By localizing the features and the objects, our method is capable of creating better association between the attributes and the objects and render the generation more recognizable. We also note that with the updated text encoder, SD 2.1 is able to handle compositional prompts remarkbly better than 1.4. Notice that while GLIGEN performs extremely well in the COCO setting, its performance in this experiment drastically drops. We hypothesize that this is because GLIGEN requires more complicated heuristics that represent the geometry of the scene correctly in order to generate more human recognizable objects. This performance gap reflects a fidelity-controllability tradeoff caused by the implementation of self attention control, which we discuss in the next ablation study section.

## 4.3 ABLATION STUDY

Fidelity-controllability tradeoff has been witnessed in many controllable generation methods (Meng et al., 2022; Tov et al., 2021): when the algorithm can control the generation better, i.e. the generation is more consistent with the user input, the generation quality of the images will usually decrease. In this section, we discuss the effect of cross attention control and self attention control on this tradeoff.

We select MultiDiffusion (MD) as the approach for self attention control and perform the COCO bounding box based generation with different ratios of diffusion timesteps to apply MD. To demonstrate the effect of cross attention control, we compare two settings with and without CAC applied to the timesteps without MD. Without CAC, we use standard Stable Diffusion with additional text prompts from the localization input concatenated to the caption for fair comparison. Figure 5 shows the performance with and without CAC at various MD ratios. With high self attention control, i.e. large MD ratios, the model can achieve high mAP50 scores which represent better consistency with the bounding boxes. However, as we control the self attention in more timesteps, the images become less realistic and thus the KID values drop. Meanwhile, with CAC applied, the model is able to achieve higher mAP50 scores with lower MD ratios and lower KID scores. This indicates that models with CAC has better fidelity-controllability tradeoff than the models without CAC in this task.

## 5 CONCLUSION & BROADER IMPACT STATEMENT

In this work, we propose to use cross attention control to provide pretrained text-to-image generative models better localized generation ability. Our method does not require extra training, model architecture modification, additional inference time, or other language restrictions and priors. We also investigate ways to incorporate large pretrained recognition models to evaluate the generation. We show qualitative and quantitative improvement compared to the base models. While this low-cost nature of our method can enhance the accessibility of better human controls over large generative models, we also recognize the potential risks involving copyright abuse, bias and inappropriate content creation associated with those pretrained models. We will implement safeguard to prevent disturbing generations when realising the code.

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

# A  IMPLEMENTATION DETAILS

## A.1  MODEL DETAILS

For the task of sampling from $x \in p_\theta(x|y_0)$ where $x \in \mathcal{X}, y_0 \in \mathcal{Y}^n$, the model will first sample from an isotropic Gaussian distribution $z_T \in \mathbb{R}^{C' \times H' \times W'} \sim p_\theta(z_T|y_0)$, and perform a $T$-step denoising procedure to gradually produce less noisy samples $z_{T-1} \sim p_\theta(z_{T-1}|y_0), \cdots, z_0 \sim p_\theta(z_0|y_0)$. $C', H', W'$ refer to the latent dimension of the Stable Diffusion model. After reaching timestep 0, the model will then obtain the final output image $x = \mathcal{D}_\theta(z_0)$ by mapping the resulting latent representation $z_0 \in \mathbb{R}^{C' \times H' \times W'}$ to the image space via a pretrained VQGAN (Esser et al., 2020) quantization-decoder $\mathcal{D}_\theta$.

We use the official implementations and Stable Diffusion (Rombach et al., 2021) as the backbone model for all models. Based on the open source availability, we choose Stable Diffusion 2.1 as the base model for all methods except StructureDiffusion (Feng et al., 2022) which uses Stable Diffusion 1.4, and GLIGEN (Li et al., 2023) which adds additional modules to the model. For GLIGEN, we use the open sourced model pretrained on Flickr (Plummer et al., 2015), VG (Krishna et al., 2016), Object365 (Shao et al., 2019), SBU (Ordonez et al., 2011) and CC3M (Sharma et al., 2018) dataset. For datasets with incomplete localization information, they generate pseudo captions with class names and pseudo bounding boxes with GLIP (Li* et al., 2022), which is a open-vocabulary semantic rich object detector.

We apply our cross attention control method to one model from each category in order to demonstrate the effectiveness of our method in diverse circumstances. We denote our method as CAC and integrate our method to Stable Diffusion 2.1, MultiDiffusion (Bar-Tal et al., 2023) and GLIGEN as an add-on. Notice that before applying our technique, only MultiDiffusion and GLIGEN have the ability to process location information. All images are generated at $512 \times 512$ resolution, which corresponds to $C = 3, H = 512, W = 512$. The latent space of VQGAN has dimensions $C' = 4, H' = 64, W' = 64$. Notice that $\lambda$ is a hyperparameter that one should tune in practice. For all quantitative analysis, we use $\lambda = 1$ for all the captions and $\lambda = 10$ for all the localized prompts. For experiments that incorporate CAC with MultiDiffusion, We use MD ratio of $0.4$ to report the quantitative results for both COCO and Cityscapes experiments and use $0.25$ for the CC-500 experiment. However, we do find lower MD ratios and $\lambda$ also provide satisfactory qualitative results. Details about each experiment setting are discussed in Section 4. Example codes of incorporating CAC with Stable Diffusion and MultiDiffusion are provided here.

## A.2  EVALUATION DETAILS

To evaluate the model performance, we use a variety of metrics for different aspects of the experiments. For fidelity, we use Kernel Inception Score (KID) (Bińkowski et al., 2018) between the generated images and filtered COCO validation set. KID imposes fewer assumptions on the data distributions than FID (Heusel et al., 2017), converges to the true values even with small numbers of samples and has also been widely used to evaluate fideilty of generative models (Meng et al., 2022).

For controllability, we use an YOLOv8 model Jocher et al. (2023) trained on the COCO dataset to predict the bounding boxes in the generated images and then calculate the precision (P), recall (R), mAP50 which is the mean average precision for bounding boxes that have an above 50% intersection-over-union (IoU) with the ground truth boxes, and mAP50-95 which is the mean average precision for bounding boxes that have a 50% to 95% IoU with the ground truth boxes.

Since the divergence between data distribution of Cityscapes images and the generated images is substantial according to the KID results in Table 2, semantic segmentation models that are only trained on Cityscapes will not work well in our setting due to the data distribution shift. As a result, we use Semantic Segment Anything (SSA) (Chen et al., 2023), which is a general purpose open-vocabulary model that leverages Segment Anything Model (SAM) (Kirillov et al., 2023) for semantic segmentation. We report the mean IoU (mIoU) score and mean accuracy (mACC) calculated from all classes, and all pixel accuracy (aACC).

In the compositional generation task we following Feng et al. (2022) and categorize each generation into three categories: generations that have incorrect or missing objects, generations that have the correct objects but the wrong colors, and the generations that have the correct objects with the correct

colors. Similar to Li et al. (2023); Feng et al. (2022), we choose GLIP (Li* et al., 2022), the semantic rich open-vocabulary object detector to produce object and color detection results. We report the mean percentage of GLIP detection (mGLIP) with confidence threshold from 0.6 to 0.8 with 0.05 step size. Considering the difficulty in recognizing compositional objects, we also perform human evaluation on Amazon Mechanical Turk (mTurk) to verify the automatic results. Details about the human evaluation are in Appendix C.

We use the CC-500 dataset (Feng et al., 2022) for the task of generating with compositional prompts. CC-500 consists of text prompts with the format "a ¡color 1¿ ¡object 1¿ and a ¡color 2¿ ¡object 2¿". For methods that require location inputs, we apply a simple heuristic where the method will generate the first object on the left hand side of the image and the second object on the right hand side. In other words, the first object will have a bounding box that spans the left half of the image and the second object will have one spanning the right half. All bounding boxes leave a 40-pixel margin to each border and the middle line of the image.

COCO is realised under Creative Commons Attribution 4.0 License. Cityscapes, CC-500 and GLIP are realised under MIT License. Yolov8 is realised under GNU Affero General Public License v3.0. SAM and SSA are realised under Apache License 2.0. We use the default thresholds for all metrics.

## B    ADDITIONAL RESULTS

In this section, we provide additional examples generated by all methods compared in all three major experiments we introduced in the main paper, as well as a qualitative illustration of the fidelity-controllability trade-off discussed in Section 4.3.

**Generating with COCO Bounding Boxes**    Figure 7 and 8 shows images generated by all compared methods with COCO bounding boxes. Our proposed CAC is able to enhance consistency between generated images and bounding box information and produce more recognizable objects while maintaining high fidelity.

**Generating with Cityscapes Semantic Segmentation Maps**    Figure 9 contains additional examples of generated images with Cityscapes semantic segmentation maps. Methods with CAC can generate more coherent and accurate images in comparison to methods without CAC.

**Generating with CC-500 Compositional Prompts**    Figure 10 and 11 provide extra samples based on CC-500 captions. CAC improves the compositional generation quality by producing more accurate compositional relationships between objects and attributes. We also provide the inference time evaluation results in Table 4.

**Ablation Study**    Figure 12 is an illustration of the fidelity-controllability trade-off that has been discussed in Section 4.3. Here we also choose MultiDiffusion as the method of controlling self attention in Stable Diffusion and "MD ratio" represents the proportion of initial diffusion timesteps that MultiDiffusion is applied to. As we can observe from the figure, without CAC, when very few steps have self attention control, the model cannot generate images that are consistent with the bounding boxes. As the ratio of self attention controlled steps increases, the generation becomes more faithful to the bounding box constraints but loses its fidelity. However, with CAC applied, the model is able to reach a sweet spot where the generated images still appear realistic while maintaining consistency with the bounding box information. This demonstrates the better fidelity-controllability trade-off provided by the CAC application.

**Generation with a Variety of User Inputs**    In Figure 13 and 14 we demonstrate additional examples of using our method to generate images with a variety of user inputs. This showcases the flexibility and effectiveness of CAC in different applications.

## C    HUMAN EVALUATION

Because of the challenges faced by large pretrained models in recognising compositional objects, we conduct human evaluations to verify the automatic metric results. We ask Amazon Mechanical Turk

| Method | Time ↓ |
|---|---|
| Stable Diffusion (SD) 1.4 | 10.59 |
| Stable Diffusion (SD) 2.1 | 9.77 |
| StructureDiffusion | **8.71** |
| **SD 2.1 + CAC (Ours)** | 9.35 |
| Composable Diffusion | 35.54 |
| MultiDiffusion (MD) | 29.59 |
| **MD + CAC (Ours)** | **17.78** |

Table 4: Cityscapes Inference Time.

workers to decide whether or not the generated images reflect the correct objects and colors in the caption. Figure 15 is an example of a task that a worker performs and the task instructions shown on the same page. Each HIT task contains one single choice question and we use all images generated from CC-500 dataset (Feng et al., 2022) by all methods compared in Table 3 for this experiment. The reward for each task is $0.12 US dollars. The actual average completion time per task is 55 seconds, and therefore the hourly compensation rate is $7.85 US dollars.

The full text of instructions is provided here: "This task requires color vision. You will see an image and a caption with descriptions of two objects Please select whether the image has (1) at least one object in the caption is missing (2) both objects but at least one of them has the wrong color (3) both objects with the correct colors".

For each individual task, additional instructions are also displayed: "Shown below are an image and a caption with descriptions of two objects. Please select the best option that describes the image.". The workers are shown a caption and an image with three options available for them to choose from: (1) "At least one object in the caption is missing" (2) "Both objects are in the image but at least one of them has the wrong color" or (3) "Both objects with the correct colors are in the image".

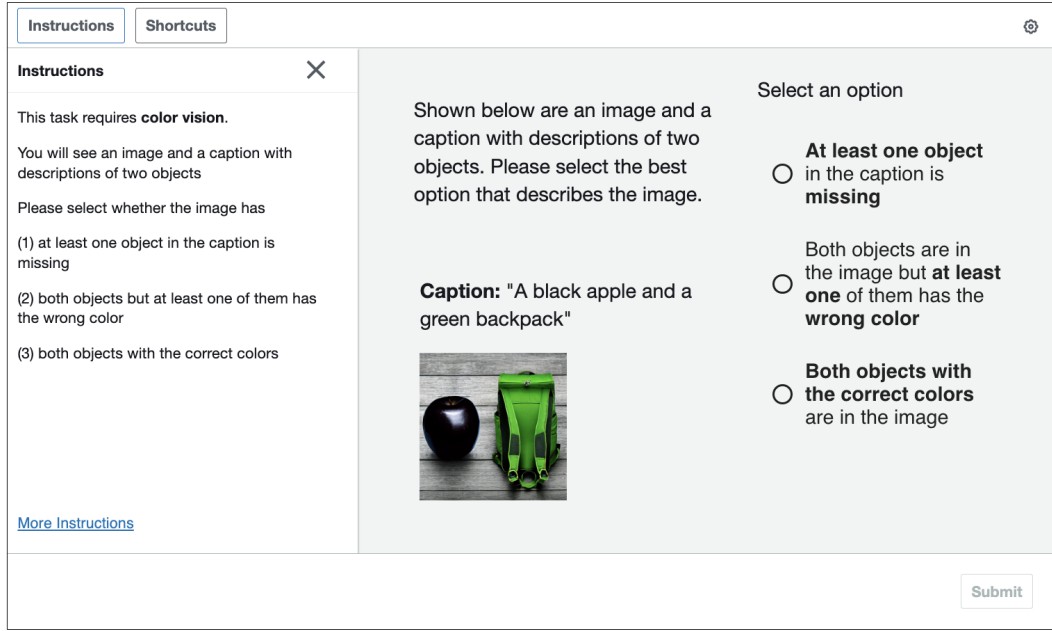

Figure 15: Example task for MTurk workers to evaluate generations with compositional prompts.

## D    EXTENDED BROADER IMPACT STATEMENT

In this section, we would like to extend the discussion of the potential societal impact of our method. As mentioned in Section 5 of the main paper, our method can provide better localized content control over large pretrained text-to-image generative models, hence has the potential of enhancing human control. CAC also adds no additional cost to the original pretrained model, allowing wide accessibility to this better controllablility.

However, we do recognize that the performance of our method is still strongly influenced by the pretrained model. As a result, our method is no exempt from the existing risks and bias observed in large pretrained text-to-image generative models. For example, without a safeguard, our method can still potentially generate adult, violent, and sexual contents similar to Stable Diffusion. Since Stable Diffusion is only trained on English captioned images, our method also has exhibits western/white dominant culture and social biases reflected in their training dataset can be reinforced in generation. Moreover, malicious users can use our method to create misinformation, discriminatory and harmful images and share copyrighted contents.

To partially mitigate these problems, we use a CLIP (Radford et al., 2021) based safety checker implemented by CompVis to filter inappropriate generations. After releasing the code, we are also committed to maintaining our open source demonstration and repository to keep up with future advancement in alleviating these issues.

## E    LIMITATIONS

As reflected in the quantitative results, CAC is by no means a perfect method for localized text-to-image generation. In this section, we discuss the limitations of our method and provide qualitative illustrations in Figure 16

The performance of our method heavily relies on the base model of choice. For example, since Stable Diffusion is only trained with English captioned images, CAC with Stable Diffusion also has limited performance on non-English prompts. Furthermore, when the base model fails to generate objects or features mentioned in the caption (as oppose to localized prompts), it is very difficult for our method to remedy that mistake. Moreover, similar to Meng et al. (2022), although usually lying in a certain small range, the optimal hyperparameters such as the MD ratios and $\lambda$ are different for individual inputs. Sub-optimal hyperparameters can compromise the quality of the generated images. In practice, we encourage users to search for the best hyperparameters for their inputs. Our method also shows weaker performance with more complicated location information where the generated images tend to look less coherent and accurate to the location information.

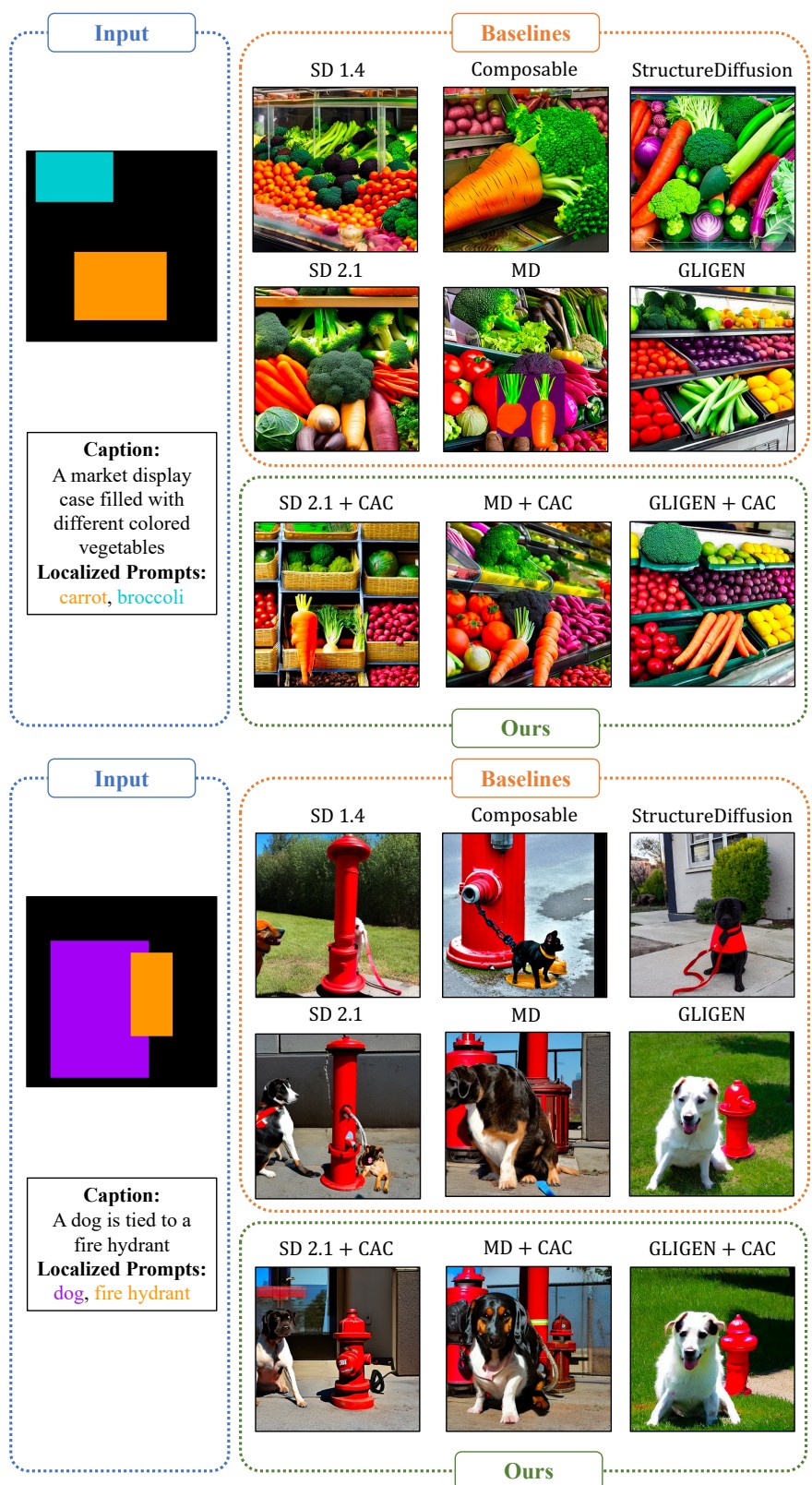

Figure 7: Additional examples of generated images based on COCO bounding boxes. "SD" denotes Stable Diffusion, "MD" denotes MultiDiffusion, and "Composable" denotes Composable Diffusion. CAC makes the generated images more recognizable and more accurate to the bounding boxes.

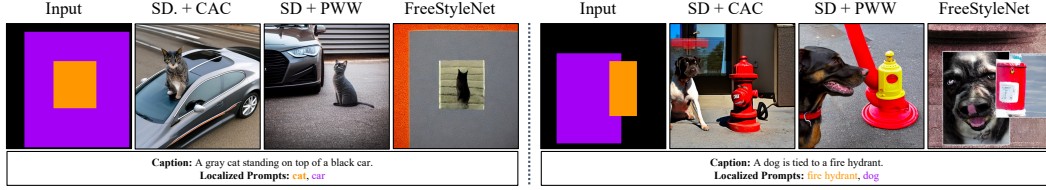

Figure 8: Additional qualitative results for generating with bounding boxes to compare our method with PWW and FreestyleNet.

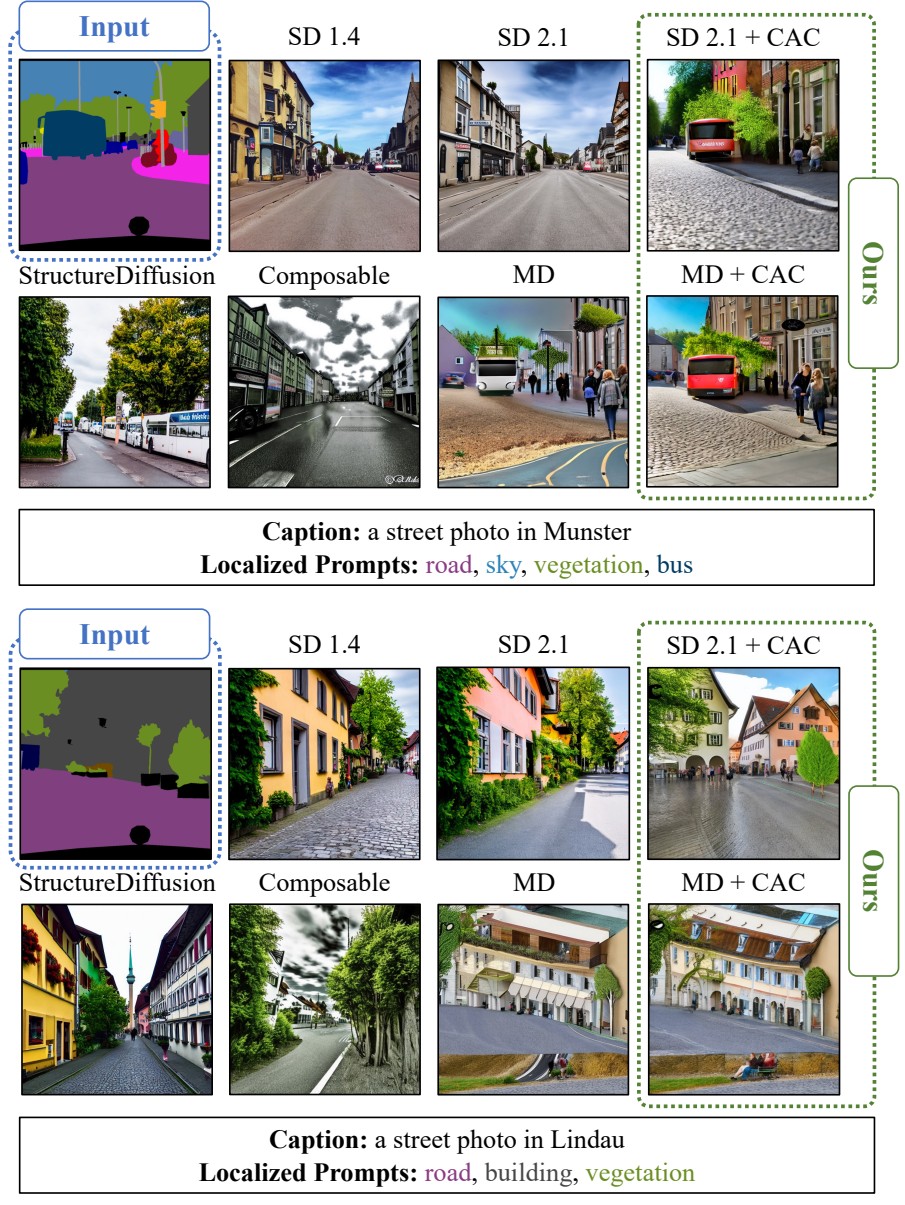

Figure 9: Additional examples of generated images based on Cityscapes semantic segmentation maps. "SD" denotes Stable Diffusion, "MD" denotes MultiDiffusion, and "Composable" denotes Composable Diffusion. Applying CAC can make the generated images more coherent while being more accurate to the semantic segmentation maps.

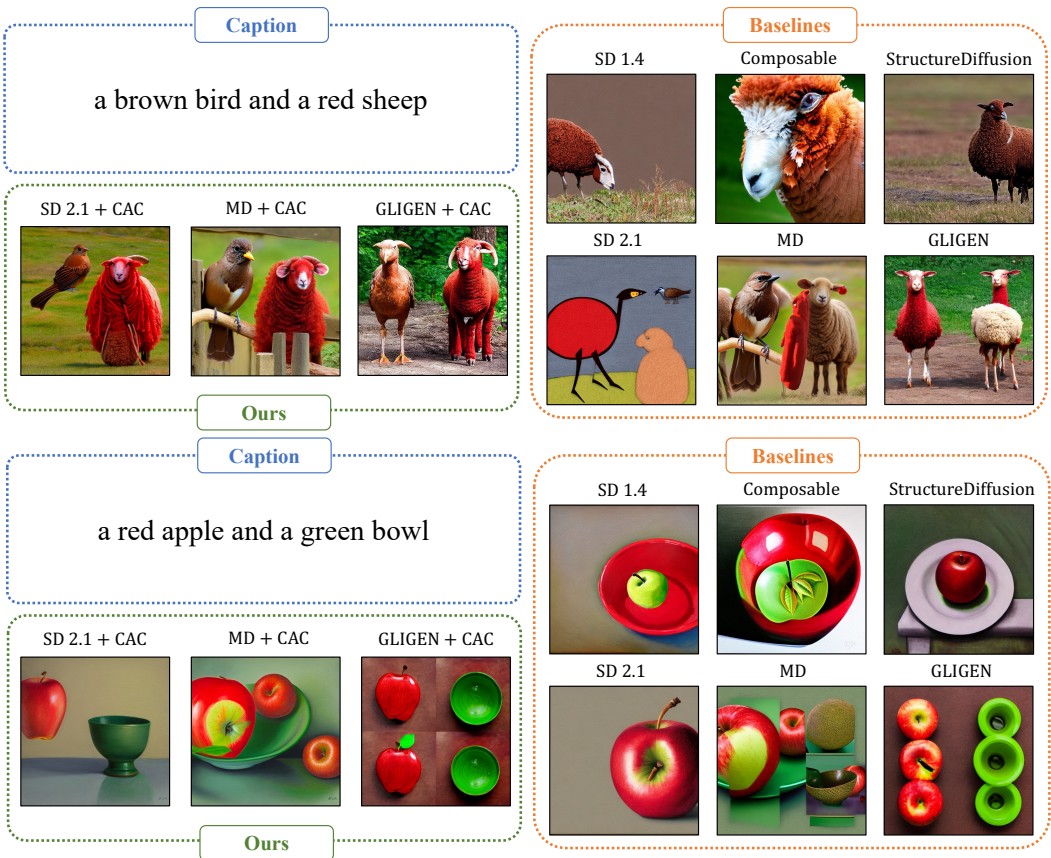

Figure 10: Additional examples of generated images based on CC-500 captions. "SD", "MD" and "Composable" denote Stable Diffusion, MultiDiffusion and Composable Diffusion. Methods with CAC generate images with more accurate compositional relationships between objects and attributes.

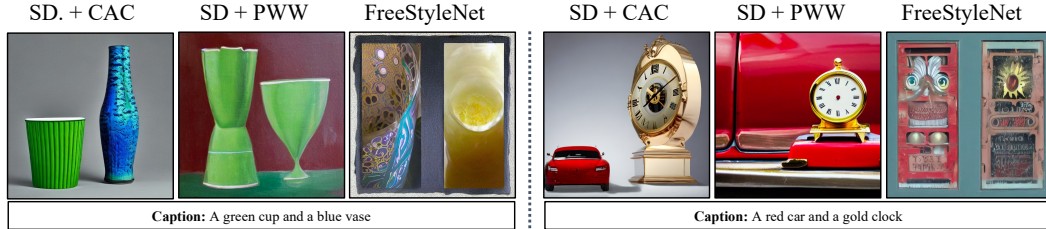

Figure 11: Additional qualitative results for generating with compositional prompts to compare our method with PWW and FreestyleNet.

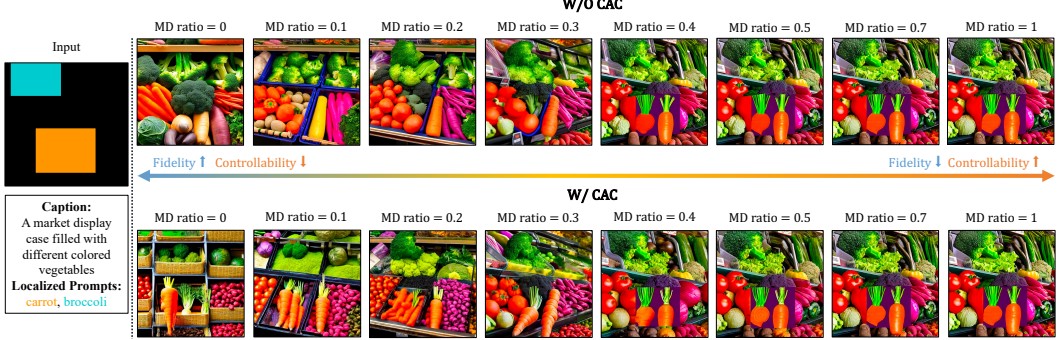

Figure 12: Comparison of the fidelity-controllability trade-offs with and without CAC. Sith CAC applied, the model is able to reach a sweet spot where the generated images appear more realistic while maintaining better consistency with the bounding box information. Samples from the same prompts with higher resolution are also shown in Figure 7.

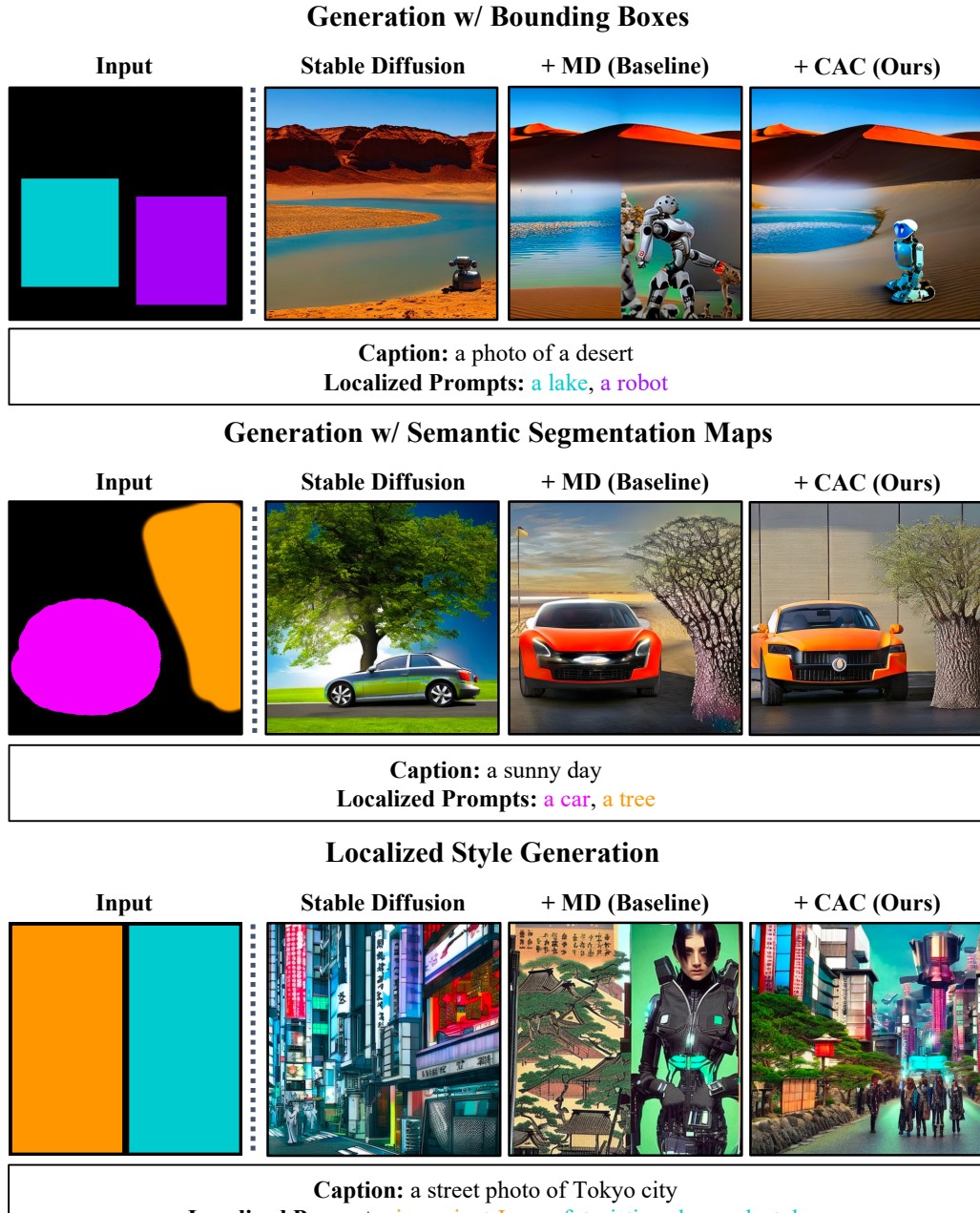

Figure 13: Examples of generated images with a variety of different types of user inputs and applications.

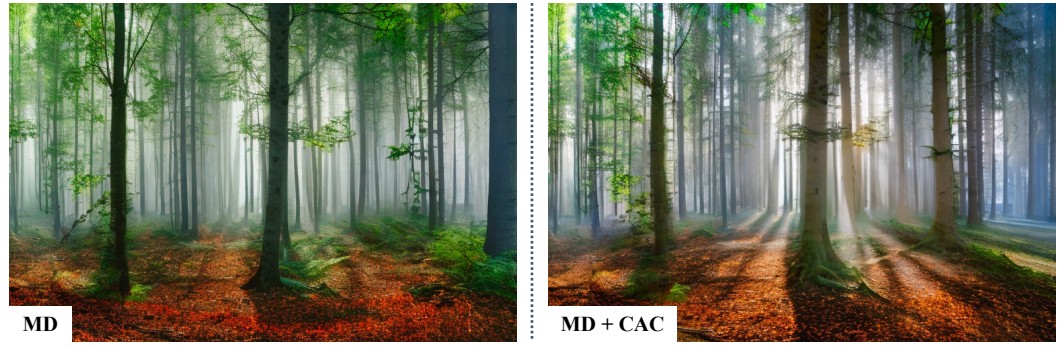

Figure 14: Panorama generation with MD and MD+CAC for "a photo of a forest". MD+CAC specifies the left part of the image to have "sunshine" and the right part to be "dark".

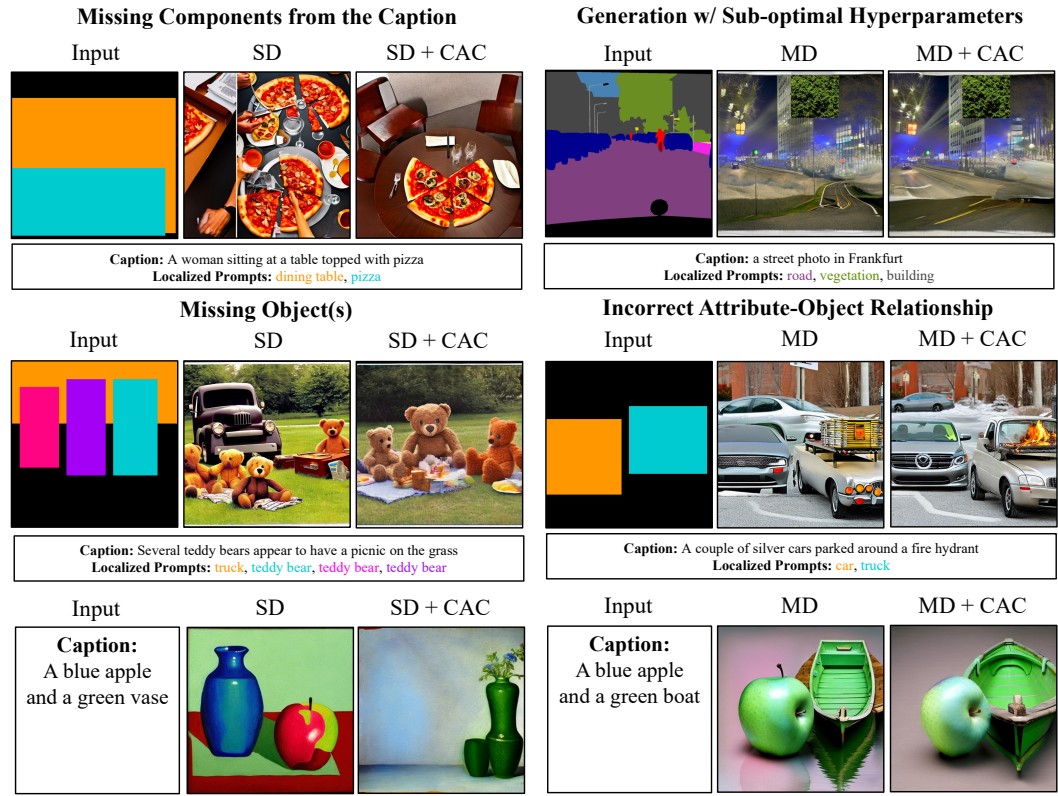

Figure 16: Example failure cases of our method. In general, the performance of our method depends of the base model and the selection of hyperparameters.

