# OpenReview forum: "Localized Text-to-Image Generation For Free via Cross Attention Control"
_ICLR.cc/2024/Conference — ICLR 2024 Conference Withdrawn Submission_

### Official Review · Reviewer_WuMR · 2023-10-28

**Soundness:** 2 fair
**Presentation:** 3 good
**Contribution:** 2 fair
**Rating:** 3
**Confidence:** 4

**Summary:**

In this paper, the authors introduce a novel layout control method called CAC (Cross-Attention Control) for text-to-image (T2I) diffusion models. CAC allows for precise control over the generated images' layout, including bounding boxes and segmentation maps, solely by manipulating the cross-attention maps during inference. Importantly, this technique is training-free and serves as an extension to existing T2I models. Experimental results demonstrate that CAC outperforms other methods like naive SD (Stable Diffusion), GLIGEN, and MultiDiffusion, showcasing its effectiveness in layout-controlled image generation.

**Strengths:**

1. This problem is of paramount importance in the field of text-to-image (T2I) diffusion models and has garnered significant research interest.

2. The proposed solution is elegantly simple. It involves controlling the cross-attention mechanism without incurring any additional computational overhead or memory usage. Moreover, it does not require any additional training.

3. The quality of the generated images appears to be quite impressive, as evidenced by the results presented in both the main paper and the supplementary image illustrations.

4. This paper is well written and easy to follow because of what the authors want to explain.

**Weaknesses:**

1. Novelty and Related Work: As I point out, the problem of layout control in text-to-image (T2I) models has received significant attention in previous research (citing references [1–14]). I would like to say that the paper lacks a thorough introduction and organization of related work in this area.

2. Comparisons and Advantages: I would like to highlight the need for comparisons between the proposed CAC method and existing methods (specifically, references [1–14]). This paper should at least address the differences and advantages of the CAC method compared to these existing approaches. Otherwise, the current version makes this papers look a bit out-of-date.


[1] Dense Text-to-Image Generation with Attention Modulation. ICCV 2023

[2] Grounded Text-to-Image Synthesis with Attention Refocusing.

[3] Zero-shot spatial layout conditioning for text-to-image diffusion models. ICCV 2023

[4] BoxDiff: Text-to-Image Synthesis with Training-Free Box-Constrained Diffusion. ICCV 2023

[5] Training-Free Layout Control with Cross-Attention Guidance.

[6] Directed Diffusion: Direct Control of Object Placement through Attention Guidance.

[7] Guided Image Synthesis via Initial Image Editing in Diffusion Model. ACM MM 2023

[8] Training-Free Location-Aware Text-to-Image Synthesis.

[9] Compositional Text-to-Image Synthesis with Attention Map Control of Diffusion Models.

[10] Controllable Text-to-Image Generation with GPT-4.

[11] Composite Diffusion | whole >= Σparts.

[12] LayoutLLM-T2I: Eliciting Layout Guidance from LLM for Text-to-Image Generation. ACM MM 2023

[13] SceneComposer: Any-Level Semantic Image Synthesis.

[14] ReCo: Region-Controlled Text-to-Image Generation.

**Questions:**

As I stated in the weakness, I would like the authors to provide a more detailed demonstration and comparison of the proposed layout control method (CAC) with existing layout control methods. It can be qualitative, quantitative, or textual comparisons with other methods to give us a better understanding of the strengths and weaknesses of CAC in relation to other approaches.

---

### Official Review · Reviewer_J2jj · 2023-10-30

**Soundness:** 3 good
**Presentation:** 2 fair
**Contribution:** 1 poor
**Rating:** 3
**Confidence:** 4

**Summary:**

This paper simply controls the cross-attention (and self-attention) maps to manipulate to have localized and coherent text-to-image generations. Because it changes the forward mechanism of the pre-trained model in samplings, it does not require any fine-tuning or architectural changes, and there is no significant inference time cost. However, it is limited to discussing novel points compared to many concurrent works with similar approaches using cross-attention manipulations [1-3] mentioned below.

**Strengths:**

- With an analysis of the cross-attention mechanism and given various localization cues, one can manipulate the generated outputs reflecting the cues.

- No significant additional inference time.

**Weaknesses:**

- There are several similar works manipulating the cross attention maps [1-3], which are not mentioned in this work. Compared with these, what points are similar and different? What would you highlight as novel contributions?

  * [1] Kim et al. (2023). Dense Text-to-Image Generation with Attention Modulation. http://arxiv.org/abs/2308.12964
  * [2] Phung et al. (2023) Grounded text-to-image synthesis with attention refocusing. http://arxiv.org/abs/2306.05427
  * [3] Chen et al. (2023) Training-free layout control with cross-attention guidance. http://arxiv.org/abs/2304.03373

**Questions:**

- In Tbl. 1, some results of GLIGEN+CAC outperform the ground truth. How do you speculate on these results regarding the validation and effectiveness of the quantitative metrics?

- In Sec. 2, Related Works -> Related Work

---

### Official Review · Reviewer_EyKK · 2023-10-31

**Soundness:** 3 good
**Presentation:** 2 fair
**Contribution:** 2 fair
**Rating:** 3
**Confidence:** 4

**Summary:**

The manuscript introduces Cross Attention Control (CAC), a novel methodology aimed at refining localized text-to-image generation. Notably, CAC enhances the precision of localized generation by proficiently maneuvering cross attention maps during the inference stage. A remarkable feature of CAC is its operational efficiency, as it necessitates no supplementary training, model architecture alterations, or additional inference time increments. Furthermore, the authors unveil a standardized assortment of evaluation metrics, leveraging substantial pre-trained models to evaluate the localized text-to-image generation.

**Strengths:**

(1) The CAC method is inspiring and easy to plug-and-play. It illuminates pathways for enhancing localized text-to-image generation without invoking the necessity for extraneous training processes or model modifications.

(2) The paper presents insightful empirical findings, demonstrating the effectiveness of CAC in improving localized generation performance with various types of location information.

**Weaknesses:**

(1) The concept of Cross-Attention Control (CAC), as depicted, follows previously explored methods, particularly resonating with well-known prompt-to-prompt methodologies. This semblance somewhat tempers the uniqueness, with the application of CAC appearing slightly surface-level without a profound analytical delve, thereby moderating the technical novelty.

(2) The terrain of localized content generation isn’t uncharted, with prior scholarly explorations such as GLIGEN, T2I-adapter, ControlNet, and UniControl leaving indelible imprints. This populated research subtly diminishes the novelty and applicational appeal of the presented work.

(3) A conspicuous omission lies in the manuscript’s comparative analysis, notably lacking in engagement with classical controllable visual generation paradigms such as T2I-adapter, ControlNet, and UniControl. This absence curtails a comprehensive evaluative perspective, considering these methodologies harbor significant functional congruences with the proposed approach.

(4) Clarity seems elusive in the methodological presentation. Illustrations such as Fig.2 seem to lack in informative richness, and the absence of detailed algorithmic diagrams or related insightful content subtly hampers the comprehension.

(5) The manuscript bears the burden of typographical errors and expressions clouded in ambiguity, casting shadows on its preparatory finesse and suitability as a stellar submission for a top-tier conference\.

**Questions:**

N/A

---

### Official Review · Reviewer_oxfo · 2023-11-02

**Soundness:** 3 good
**Presentation:** 3 good
**Contribution:** 3 good
**Rating:** 8
**Confidence:** 4

**Summary:**

The paper addresses a very important problem in T2I: controllability in terms of generating specific elements within the image. The key idea of the paper is to show that localized generation can be achieved by controlling cross attention maps during inference. A very attractive aspect of the proposed framework is that it does not require extra training, model architecture modification and additional inference time. Qualitative and quantitative evaluations demonstrate the utility of the proposed approach.

**Strengths:**

The paper is very interesting and does a great job in terms of comparing the proposed approach with prior works. The results look great and the evaluations are solid. Overall, an impressive work that could have diverse applications in Gen AI applications.

**Weaknesses:**

I do not find any major weaknesses with this work. I have a few questions: (1) How does the proposed approach compares to approaches like LayoutGPT in terms of quality. LayoutGPT[1] also doesn’t require any additional training and leverages the GLIGEN model. (2) The work mainly leverages models like SD 1.4 and SD 2.1. Will some of the issues spotted in this paper applicable to more recent models like SDXL 1.0? (3) In Figure 1, it was mentioned that the proposed method is applicable to localized style generation, but I didn’t find any discussion/results/evaluation on this?

1. Feng, W., Zhu, W., Fu, T. J., Jampani, V., Akula, A., He, X., ... & Wang, W. Y. (2023). LayoutGPT: Compositional Visual Planning and Generation with Large Language Models. arXiv preprint arXiv:2305.15393.

**Questions:**

Please see Weaknesses